# Angstrom-confined catalytic water purification within Co-TiO$_x$ laminar membrane nanochannels

Chenchen Meng[1,2,8], Baofu Ding[3,4,8], Shaoze Zhang[5,8], Lele Cui[1], Kostya Ken Ostrikov[6], Ziyang Huang[4], Bo Yang [2], Jae-Hong Kim [7] & Zhenghua Zhang [1✉]

The freshwater scarcity and inadequate access to clean water globally have rallied tremendous efforts in developing robust technologies for water purification and decontamination, and heterogeneous catalysis is a highly-promising solution. Sub-nanometer-confined reaction is the ultimate frontier of catalytic chemistry, yet it is challenging to form the angstrom channels with distributed atomic catalytic centers within, and to match the internal mass transfer and the reactive species' lifetimes. Here, we resolve these issues by applying the concept of the angstrom-confined catalytic water contaminant degradation to achieve unprecedented reaction rates within 4.6 Å channels of two-dimensional laminate membrane assembled from monolayer cobalt-doped titanium oxide nanosheets. The demonstrated degradation rate constant of the target pollutant ranitidine (1.06 ms$^{-1}$) is 5–7 orders of magnitude faster compared with the state-of-the-art, achieving the 100% degradation over 100 h continuous operation. This approach is also ~100% effective against diverse water contaminates with a retention time of <30 ms, and the strategy developed can be also extended to other two-dimensional material-assembled membranes. This work paves the way towards the generic angstrom-confined catalysis and unravels the importance of utilizing angstrom-confinement strategy in the design of efficient catalysts for water purification.

[1] Institute of Environmental Engineering & Nano-Technology, Institute of Environment and Ecology, Guangdong Provincial Engineering Research Centre for Urban Water Recycling and Environmental Safety, Tsinghua Shenzhen International Graduate School, Tsinghua University, Shenzhen 518055, China. [2] College of Chemistry and Environmental Engineering, Shenzhen University, Shenzhen 518060, China. [3] Institute of Technology for Carbon Neutrality/Faculty of Materials Science and Engineering, Shenzhen Institute of Advanced Technology, CAS, Shenzhen 518055, China. [4] Shenzhen Geim Graphene Center, Tsinghua-Berkeley Shenzhen Institute and Tsinghua Shenzhen International Graduate School, Tsinghua University, Shenzhen 518055, China. [5] National Engineering Laboratory for Vacuum Metallurgy, Engineering Laboratory for Advanced Battery and Materials of Yunnan Province, Kunming University of Science and Technology, Kunming 650093, China. [6] School of Chemistry and Physics and QUT Centre for Materials Science, Queensland University of Technology (QUT), Brisbane 4000, Australia. [7] Department of Chemical and Environmental Engineering, Yale University, New Haven, Connecticut 06511, USA. [8] These authors contributed equally: Chenchen Meng, Baofu Ding, Shaoze Zhang. ✉email: zhenghua.zhang@sz.tsinghua.edu.cn

Leveraging nanoconfinement effects in catalysis has attracted tremendous attention due to the ability to enhance intrinsic characteristics of a catalytic system, *e.g.*, electronic state, mass and electron transfer, phase behavior, and reaction rates[1,2]. By confining catalytic reactions within a nanoscale space, the selectivity can be significantly enhanced and the reaction kinetics can be massively accelerated under mild process conditions[2–4]. Consequently, nanoconfinement catalysts with unique geometric and electronic structures have been developed to enhance the catalytic performance for water treatment[5–9]. For instance, $Fe_2O_3$ nanoparticles (~2 nm) confined within multi-walled carbon nanotube (CNT) with an inner diameter of 7 nm show 22.5 times higher methylene blue degradation rates than without the CNT encapsulation[8]. As another example, iron oxychloride (FeOCl) catalysts loaded within the pores (~20 nm) of a ceramic ultra-filtration membrane increased the degradation rates of para-chlorobenzoic acid (pCBA) by 3 orders of magnitude compared with the batch suspension reaction[9].

Catalyst confinement at angstrom-scales has been theoretically-predicted to have much better catalytic performance owing to the stronger electronic interactions, spatial localization, and more effective diffusion and reactions than in the nanometer-confinement case[2,10]. For example, density functional theory (DFT) calculations indicate that a much higher catalytic activity of CNT filled with metal or metal oxides could be achieved when the inner diameter of CNT is reduced to 4–6 Å compared to CNTs thicker than 1.0 nm[10]. Even though it is attractive to implement angstrom-confinement catalysis (ACC) for water treatment, several challenges such as angstrom-scale channel formation, non-uniform distribution and activity of catalytic centers at angstrom-scale, prevent the ACC practical implementation[11–13].

Two-dimensional (2D) materials have atomic-level thickness, and the laminar membranes assembled from 2D materials possess multi-layer stacked structures and can facilitate the transport of small molecules through the interlayer nanochannels[14–16]. Furthermore, the large capillary-like force (>50 bar) generated within the nanochannels can substantially accelerate molecular transport[17,18]. Therefore, the laminar membranes assembled from 2D materials can be ideal for the angstrom-confined catalytic water treatment. However, the effect of the interlayer spacing of 2D laminar membranes on angstrom-confinement catalysis remains essentially unknown. Moreover, the currently available 2D laminar membranes lack catalytic centers within the interlayer space. Here, for the first time, we implement the generic angstrom-confined catalytic water contaminant degradation concept within 4.6 Å channels of 2D laminate membrane assembled from monolayer cobalt-doped titanium oxide (Co-TiO$_x$) nanosheets and demonstrate the long-term catalytic performance with unprecedentedly-high pollutant degradation kinetics.

Our approach is based on peroxymonosulfate (PMS)-based advanced oxidation processes (AOPs) which have drawn tremendous attention in water and wastewater treatment given their higher reactivity, higher yield of radical formation, and higher redox potential ($E^0(SO_4^{\bullet-}/SO_4^{2-}) = 2.6-3.1\,V > E^0(^{\bullet}OH/{}^{-}OH) = 1.9-2.7\,V$) compared to $H_2O_2$-based AOPs[19]. Among all transition metals, Co is the most effective for PMS activation for the generation of $SO_4^{\bullet-}$ and $^{\bullet}OH$ radicals[20]. Titania nanosheets possess good chemical stability, atomically-thin monolayer structure, and high specific surface area, while the doping of cobalt ions into the titania lattice can introduce active sites for the activation of PMS to achieve good recyclability and stability[6,21,22]. Meanwhile, the numerous angstrom-confined channels within the Co-TiO$_x$ membrane can act as catalytic reactors to significantly expose the catalytic active sites, to accelerate the mass transfer of reactants and to produce reactive

oxygen species (ROS) during the filtration process. Such angstrom-confined catalysis was realized by activating PMS within the angstrom-sized interlayer spaces of the membrane (Fig. 1a).

First, the adsorption energy ($E_{ads}$) of PMS molecule intercalating between the nanosheets of Co-TiO$_x$ membrane with the variable interlayer free spacing was calculated using DFT (Fig. 1b). Interestingly, with the decrease of the interlayer free spacing, three different zones can be clearly distinguished:

(i)   A steady zone between 50 Å to 10 Å, whereby a stable $E_{ads}$ (−0.82 to −0.91 eV) was calculated. This result indicates that PMS is weakly adsorbed onto the Co-TiO$_x$ nanosheets with the large interlayer free spacings.

(ii)  A transitional zone between 10 Å to 5.9 Å, whereby an increasing $E_{ads}$ (−0.91 to −1.27 eV) with the decreasing interlayer spacing was calculated. This result indicates that PMS is more strongly adsorbed onto the Co-TiO$_x$ nanosheets with the narrow interlayer spacings due to the additional interactions (e.g., electrostatic, van der Waals, and orbital interactions) from the upper layer.

(iii) A transient zone with an interlayer free spacing of ≤5.8 Å, whereby a remarkably high $E_{ads}$ (−3.53 to −4.06 eV) was calculated. In this case, PMS is dissociated through the cleavage of its S–O bond.

In this work, guided by these theoretical insights, we synthesize an angstrom-confined catalytic membrane by stacking Co-TiO$_x$ nanosheets with an interlayer spacing of <5.8 Å and demonstrate unprecedented reaction rates of water contaminant degradation within 4.6 Å channels of two-dimensional Co-TiO$_x$ laminate membrane.

## Results and discussion

**Synthesis and microanalysis of the Co-TiO$_x$ membrane.** Lepidocrocite-type 2D Co-TiO$_x$ nanosheets were synthesized using the simple annealing, protonation, ion exchange, and exfoliation steps (Fig. S1, S2). The average thickness (1.2 nm) and lateral size (1.7 μm) of the lamellar Co-TiO$_x$ nanosheets were confirmed by the atomic force microscopy (AFM) measurements (Fig. 2a and Fig. S3). Considering the thickness of a 2D titania single-crystal nanosheet (~0.75 nm)[23,24] and the diameter of the adsorbed molecules (such as $H_2O$), the as-prepared Co-TiO$_x$ consists of monolayer nanosheets. Transmission electron microscopy (TEM) image (Fig. 2b) displays an ultrathin, and nearly transparent Co-TiO$_x$ nanosheets with an excellent dispersibility (Fig. 2b inset). Energy-dispersive X-ray spectroscopy (EDX) elemental mapping of the Co-TiO$_x$ nanosheets suggests that Ti, O, and Co were uniformly distributed across the sample, confirming the successful synthesis of Co-TiO$_x$ nanosheets (Fig. S4). X-ray photoelectron spectroscopy (XPS) was performed to identify the valence state of Co-ions (Fig. S5) and confirms the partial substitution of ≡Co(II) (76 at%) and ≡Co(III) (24 at%) ions for ≡Ti(IV) ions at the octahedral sites of Co-TiO$_x$ with the formation of Co-TiO$_x$.

The high-resolution TEM (HRTEM) image (Fig. 2c) reveals two lattice spacings of 0.32 nm and 0.38 nm, which correspond to the basal planes (110) of lepidocrocite-type TiO$_x$ and Co-TiO$_x$, respectively. X-ray diffraction (XRD) patterns of the parent layered TiO$_x$ and Co-TiO$_x$ demonstrate sharp diffraction peaks, which indicates that both samples possess high crystallinity (Fig. 2d). The typical diffraction peaks of TiO$_x$ matched well to those of a lepidocrocite-type crystal structure (Fig. S6). After doping TiO$_x$ with Co, there was an increase in the $d$-spacing from 3.67 Å to 3.76 Å as shown by the slight shift of the peak (110) position from 24.2° to 23.6° (Fig. 2d inset). Such a result obtained

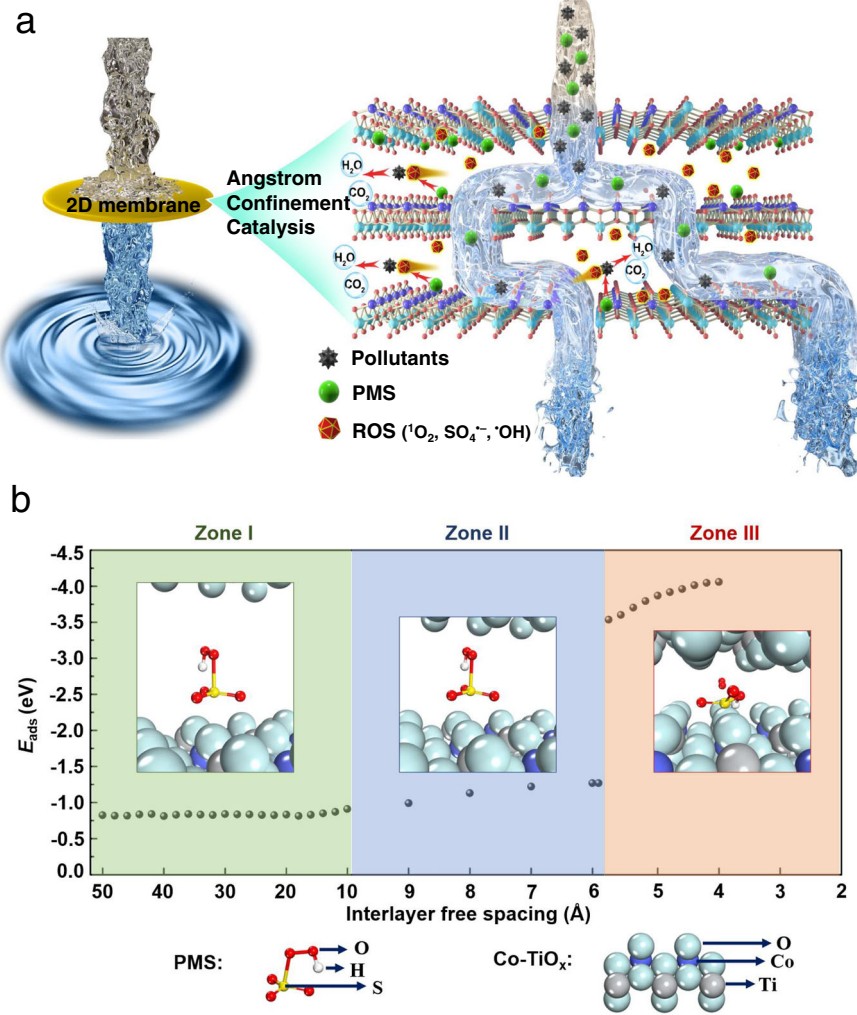

**Fig. 1 Atomistic design of effective angstrom-confinement catalysis. a** Schematic diagram of ACC in 2D laminate membrane. **b** Adsorption energy ($E_{ads}$) of PMS molecule intercalating into the nanosheets of Co-TiO$_x$ membrane with the varied interlayer free spacing.

from XRD is consistent with the lattice size determined from HRTEM analysis (Fig. 2c). The increased $d$-spacing induced by Co dopants confirms the elongation of the unit cell along $a$ and $c$-axes as larger ≡Co(II) ions (81 pm) replaced the smaller ≡Ti(IV) ions (74 pm) in the Co-TiO$_x$ lattice[25].

We then fabricated the Co-TiO$_x$ membrane by subjecting Co-TiO$_x$ nanosheets to vacuum-assisted filtration on a mixed cellulose ester membrane (MCE) support (Fig. 2e inset). As shown in Fig. 2e, the surface of Co-TiO$_x$ membrane was relatively smooth with a small degree of wrinkling, but no defects, pinholes, or cracks were observed in the SEM image. The EDX mapping (Fig. 2e and Fig. S7) reveals the uniform distributions of Co, Ti, and O elements across the Co-TiO$_x$ membrane with an atomic Co: Ti: O ratio of 0.4: 1.6: 4, which is consistent with the theoretical atomic ratio. The cross-sectional SEM image (Fig. 2f) indicates that the membrane was ~500 nm thick, with a distinct laminated structure.

The XRD patterns (Fig. 2g) of Co-TiO$_x$ membrane in both dried and hydrated states exhibit sharp (020) diffraction peaks, which indicates that the Co-TiO$_x$ membrane possessed a highly-ordered stacking arrangement. Using the Bragg's law, the $d$-spacing of the dried Co-TiO$_x$ membrane was calculated to be 1.05 nm and this increased to 1.15 nm for the hydrated membrane. The thickness of the lamellar Co-TiO$_x$ nanosheet was also estimated using the cross-sectional HRTEM image of a

section of the sonicated Co-TiO$_x$ membrane (Fig. 2h). The alternating dark-and-light stripe patterns represent the interlayer free spacing and the Co-TiO$_x$ nanosheets, respectively. The Co-TiO$_x$ cross-sectional HRTEM image demonstrates the average lamellar thickness of ~0.69 nm, which matches the theoretical thickness of a unilamellar TiO$_2$ nanosheet (~0.70–0.75 nm)[23,24]. Thus, both XRD and HRTEM results collectively suggest that the interlayer free spacing for effective angstrom-confined catalytic space in dried and hydrated states is ~3.6 and ~4.6 Å, respectively. Hence, the as-prepared Co-TiO$_x$ membrane with angstrom-confined spaces of ~4.6 Å in its hydrated state (Fig. 2i) was used in the subsequent catalytic experiments.

**Angstrom-confinement catalysis performance**. The angstrom-confinement catalysis performance of Co-TiO$_x$ membrane by activating PMS through numerous angstrom-confined membrane interlayer channels was then evaluated using ranitidine as a target pollutant (Fig. 3a). Firstly, the thickness of Co-TiO$_x$ membrane was optimized in terms of its catalytic performance and water flux. With the increasing membrane thickness, the membrane permeance decreased while the removal efficiency increased, whereby the 100% removal efficiency was achieved at a membrane thickness of ≥ 500 nm. Therefore, Co-TiO$_x$ membrane with a thickness of 500 nm, corresponding to the Co-TiO$_x$ loading of 0.08 mg/cm$^2$, was selected since the 100% removal efficiency and

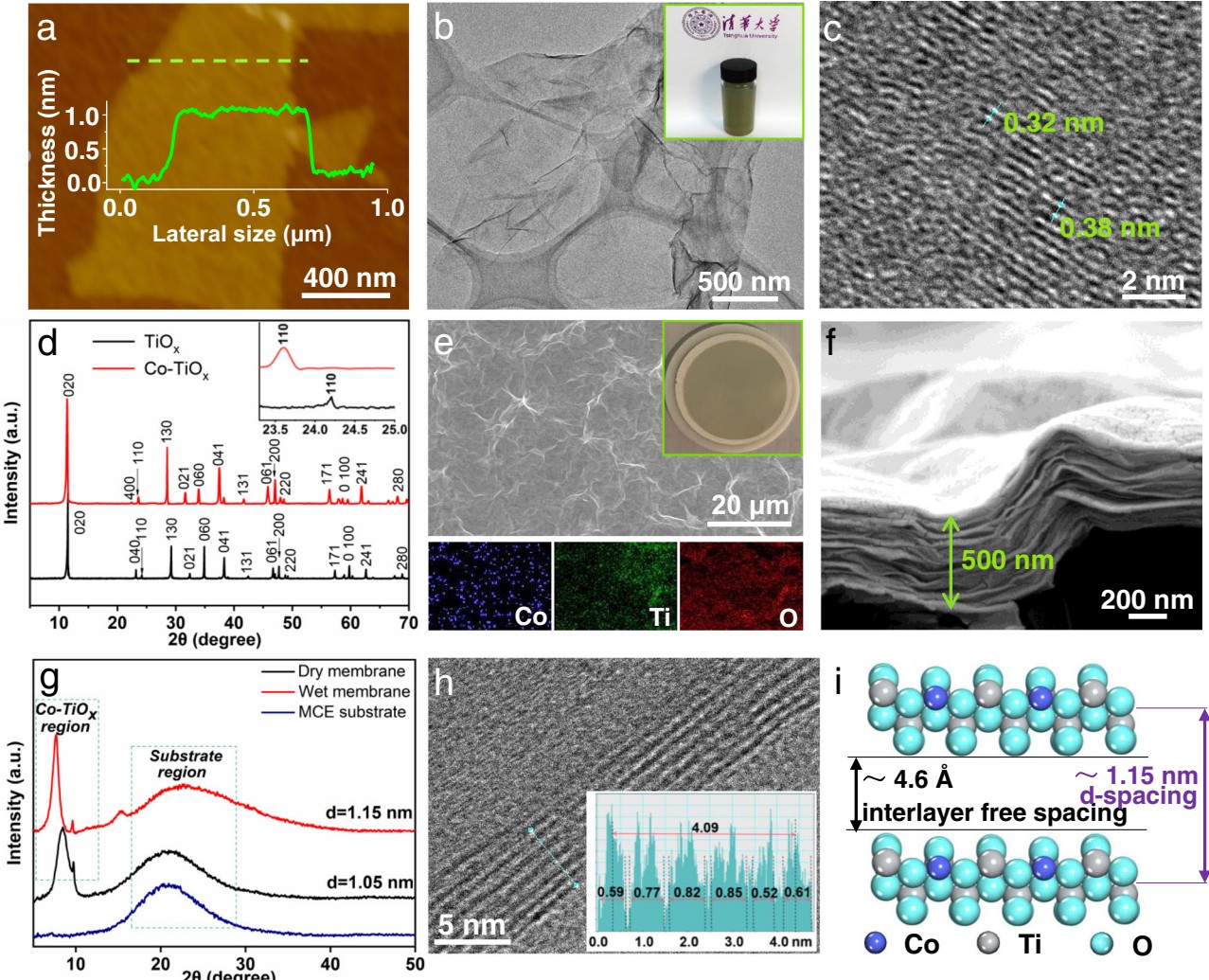

**Fig. 2 Synthesis and microanalysis of the Co-TiO$_x$ membrane. a** AFM image and corresponding height profile (green dashed line), **b** TEM image and (inset) corresponding aqueous colloidal solution, and **c** HRTEM image of Co-TiO$_x$ nanosheets. **d** XRD patterns of parent layered TiO$_x$ and Co-TiO$_x$ powders, and (inset) the magnified XRD patterns at low Bragg angles. **e** SEM image and EDX mapping of Co-TiO$_x$ membrane, and (inset) photograph of Co-TiO$_x$ membrane on MCE support. **f** Cross-sectional SEM image. **g** XRD patterns in both dried and hydrated states of Co-TiO$_x$ membrane. **h** Cross-sectional HRTEM image, and (inset) grayscale profile of partial Co-TiO$_x$ membrane sonicated. **i** Illustration of the interlayer free spacing and *d*-spacing between neighboring Co-TiO$_x$ nanosheets in the membrane (hydrated state).

permeance of 131 L·m$^{-2}$·h$^{-1}$ were considered to be the optimal ACC conditions. The molecular size and adsorption energy for PMS, ranitidine and water intercalated into Co-TiO$_x$ nanosheets with an interlayer free spacing of 4.6 Å were calculated by DFT (Fig. S8). The molecular sizes of PMS, ranitidine and water are 1.078, 2.659 and 0.627 Å, respectively, and the adsorption energy follows the order: ranitidine ($-9.416$ eV) > PMS ($-4.119$ eV) > water ($-2.070$ eV). These results indicate that ranitidine would adsorb onto the Co-TiO$_x$ nanosheets more preferentially than PMS and water during the transport through the 4.6 Å interlayer free spacings of the Co-TiO$_x$ membrane.

We further investigated the effects of Co-TiO$_x$ membrane and PMS on the angstrom-confined catalysis of ranitidine (Fig. 3b). Using only PMS without the catalyst exerted negligible effects on the ranitidine removal since PMS was unable to generate sufficient active radicals *via* self-decomposition[19]. In the absence of PMS, both Co-TiO$_x$ and TiO$_x$ membranes exhibited also very poor ranitidine removal performances (<20%) as they merely relied on the molecular sieving and adsorption mechanism. As such, the removal efficiency decreased over time, and further

ranitidine removal eventually ceased after 30 min due to the adsorption saturation in the membrane. In addition, TiO$_x$ membrane without Co dopants also exhibited a low ranitidine removal efficiency of 20.3% within 30 min. This result suggests the importance of Co ion sites in Co-TiO$_x$ lattice for PMS activation and catalytic reactions. Consistently, a suspension of Co-TiO$_x$ nanosheets and PMS achieved a much higher ranitidine removal efficiency of 85.4% within 20 min.

In marked contrast, the ACC enabled by the Co-TiO$_x$ layered membrane produced the extremely short calculated retention time of only 5.5 ms (Fig. S9) for the 100% ranitidine removal (Fig. 3c). Moreover, the first-order rate constant of the ACC ranitidine degradation process was 63600 min$^{-1}$ (1.06 ms$^{-1}$) (inset Fig. 3c), which is about 6 orders of magnitude faster than that for the non-confined heterogeneous Co-TiO$_x$ nanosheets suspension (0.092 min$^{-1}$) (Fig. S10). Such a rate (63600 min$^{-1}$) is also 5–7 orders of magnitude faster than that achieved by other means (0.0032–0.33 min$^{-1}$) (Fig. 3d and Table S1). Indeed, these results imply that designing the Co-TiO$_x$ membrane with the angstrom-scale spatial confinement can significantly enhance the

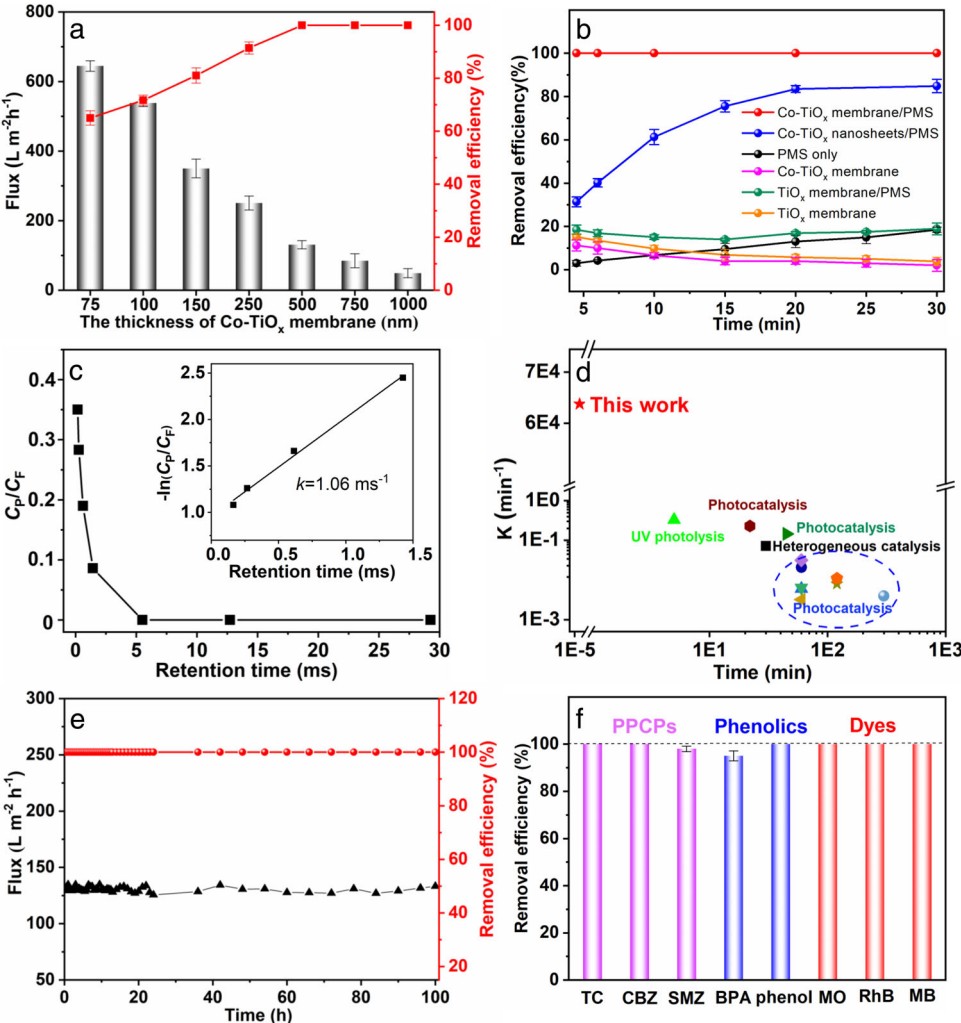

**Fig. 3 Angstrom-confinement catalysis performance of the Co-TiO$_x$ membrane. a** Permeance of Co-TiO$_x$ membranes and efficiency of ranitidine removal as a function of membrane thickness. **b** Removal efficiency of ranitidine in different reaction systems. **c** Normalized concentration of ranitidine ($C_{permeate}$/$C_{feed}$) versus membrane retention time. The inset showing the first-order kinetics. **d** Comparison of the first-order rate constant $k$ values. **e** Stability test of flux and removal efficiency with operation duration. **f** Removal efficiencies of PPCPs, phenolics, and dyes. Conditions of the feed solution for all membrane tests: [Ranitidine] or [Other pollutants] = 5 mg/L, [PMS] = 0.16 mM, and pH 4.0. Error bars in **a**, **b** and **f** represent standard deviation of three measurements on the same membrane.

performance compared to common catalytic systems. The achieved high catalytic activity towards ranitidine can be attributed to the angstrom-confinement effect of the Co-TiO$_x$ membrane.

The stability of Co-TiO$_x$ membrane was further investigated in the pressure-dependent continuous flow experiments. The flux and the degradation efficiency of ranitidine remained nearly unchanged with an operation duration of up to 100 h (Fig. 3e). Moreover, the amount of Co leached during the experiment was 0.2 µg/L (Table S2), which is far below the World Health Organization (WHO) benchmark for drinking water quality, i.e., 10 µg/L. This superior removal performance and stability can be primarily ascribed to the stable and efficient lattice-doped active sites within the lamellar Co-TiO$_x$ nanosheets. To evaluate the chemical stability of Co-TiO$_x$ membrane, XPS analysis was conducted before and after the stability test (Fig. S5). Less than 10% ≡Co(II) was converted into ≡Co(III) after the 100 h stability test (Table S3). The Co contents in Co-TiO$_x$ membranes before and after the stability test remained nearly unchanged, which further suggests the excellent stability of the catalytic sites in the Co-TiO$_x$ lattice.

To verify the generic applicability of the Co-TiO$_x$ membrane-based angstrom-confined catalysis, the removal performances of multiple pollutants such as pharmaceutical and personal care products (PPCPs) (tetracycline hydrochloride (TC), carbamazepine (CBZ), sulfamethizole (SMZ)), phenols (bisphenol A (BPA), phenol), and dyes (methyl orange (MO), rhodamine B (RhB), and methylene blue (MB)) were further investigated. As shown in Fig. 3f, all pollutants were rapidly removed with a retention time of <30 ms. This result is 3 orders of magnitude faster than that achieved by the FeOCl membrane-based nanoconfinement catalysis (~20 nm) while using two orders of magnitude less catalyst (1 mg here versus 210 mg of FeOCl)[9]. In comparison, the PMS system without the catalyst exerted negligible effects on the removal of the above-mentioned pollutants (Fig. S11). Therefore, it can be concluded that the Co-TiO$_x$ membrane-based ACC is a robust AOP with potentially generic applicability.

**Mechanism of angstrom-confinement catalysis.** Towards understanding the angstrom-confinement catalysis, enrichment of reactants, more effective diffusion and reactions, and stronger

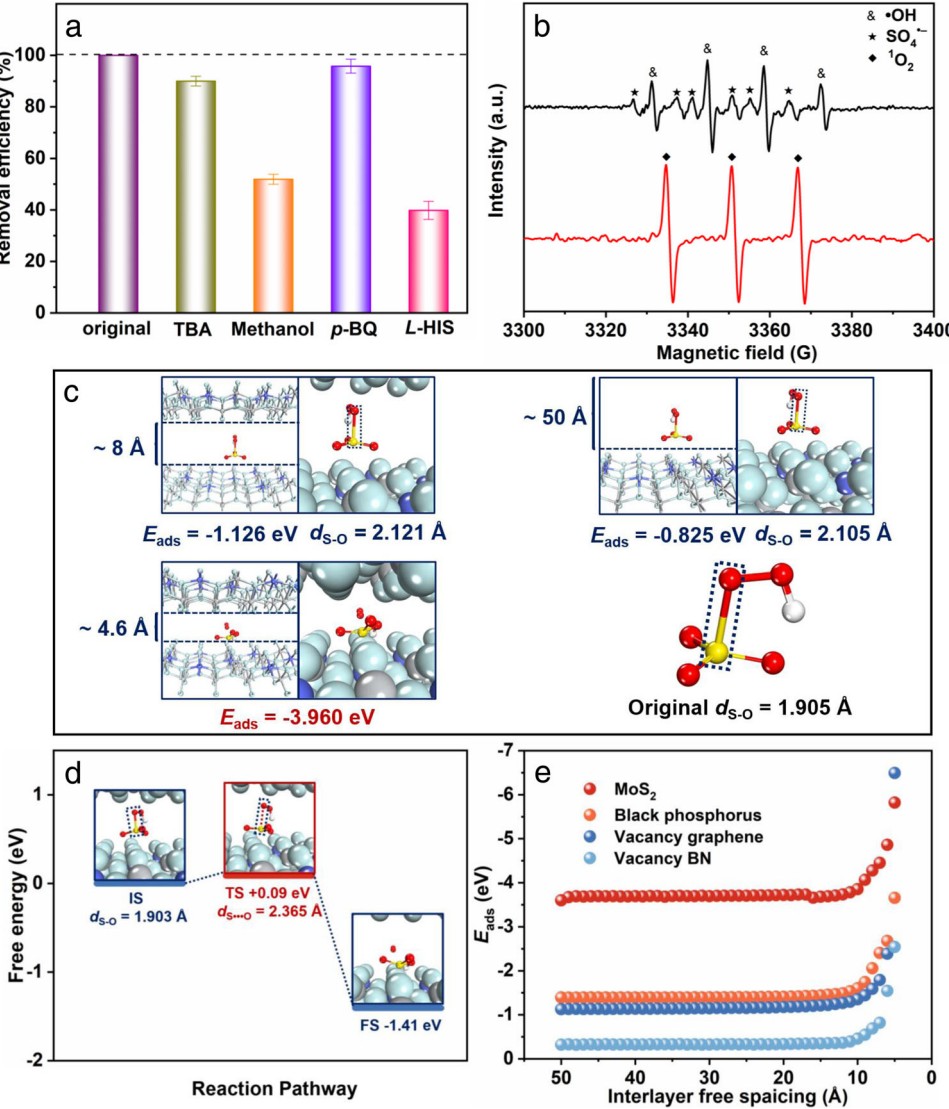

**Fig. 4 Mechanism of angstrom-confinement catalysis performance of the Co-TiOₓ membrane and applicability to diverse material systems.**
**a** Comparison of the removal efficiency under different quenching conditions. Error bars represent standard deviation of three measurements of quenching experiment. **b** EPR spectra of Co-TiOₓ membrane/PMS system. Reaction condition: [ranitidine] = 5 mg/L, [PMS] = 0.16 mM, [DMPO] = 0.35 mM, [TEMP] = 25 mM, initial solution pH = 4.0, and T = 298 K. **c** Energetic and geometric data of the models with representative interlayer free spacings. **d** Energy profile during the dissociation of PMS within Co-TiOₓ membrane. **e** Adsorption energies of PMS molecule intercalating in other 2D nanosheet-assembled membranes with the varied interlayer free spacing.

electronic interactions at angstrom-scales have been theoretically-proposed to be the potential main contributors to the enhanced catalytic performance[2,10]. As such, the mechanism of the Co-TiOₓ membrane-based angstrom-confined catalysis activated with PMS was investigated using ROS scavenging experiments, electron paramagnetic resonance (EPR), DFT, ab initio molecular dynamics (AIMD) and molecular dynamics (MD).

First, a series of selective radical quenching tests were performed to determine the predominant reactive species in the Co-TiOₓ membrane/PMS system (Fig. 4a and Fig. S12). Radical scavengers, i.e., tert-butanol (TBA, for •OH) and methanol (for both •OH and SO₄•⁻), were added into the system separately. Based on the results, the systems with tert-butanol and methanol exhibited the reduced ranitidine degradation efficiency by 10.0% and 48.1%, respectively. When p-benzoquinone (p-BQ, a scavenger for •O₂⁻) was added into the reaction system, ranitidine degradation efficiency was slightly reduced by ~4%, which may be attributed to the obstruction of the active sites by

the adsorbed p-benzoquinone. In addition, with the addition of L-histidine (L-HIS) into the reaction system as a scavenger for singlet oxygen ($^1O_2$), the ranitidine degradation efficiency was significantly reduced to 39.6%. These results suggest that the decomposition of ranitidine mainly occurs by reacting with $^1O_2$, SO₄•⁻ and •OH generated during the PMS activation.

EPR experiments further verified the dominant active species in the Co-TiOₓ membrane/PMS system. 2,2,6,6-tetramethyl-4-piperidone hydrochloride (TEMP) was used to trap singlet oxygen ($^1O_2$) species, and 5,5-dimethyl-1-pyrroline (DMPO) to trap •OH, SO₄•⁻, and •O₂⁻. As shown in Fig. 4b, the characteristic signals for both DMPO−•OH (Aₙ = Aₕ = 14.9 G) and DMPO−SO₄•⁻ (Aₙ = 13.2 G, Aₕ = 9.6 G, Aₕ = 1.48 G, and Aₕ = 0.78 G) adducts were detected after 2 min, which indicates the production of •OH and SO₄•⁻ in the Co-TiOₓ membrane/PMS system. A triplet peak signal (1:1:1, Aₙ = 16.9 G) was also observed, which indicates the existence of $^1O_2$ species in the system. The DMPO−•O₂⁻ signal was undetected in the

system, which is consistent with the radical quenching results and this further confirms the absence of $^•O_2^-$ in the system. Moreover, the variation in the intensity of EPR signals with filtration duration (Fig. S13) reveals the initial increase in the signal intensities of $^•OH$ and $^1O_2$, and these intensities remained constant during the test. In contrast, $SO_4^{•-}$ signal decreased with reaction duration since $SO_4^{•-}$ can capture $H_2O$ to generate $^•OH$ radicals[26]. Therefore, among all the identified active species, $^1O_2$, $SO_4^{•-}$ and $^•OH$ play key roles in the Co-TiO$_x$ membrane-based angstrom-confined catalytic degradation process. Compared with the batch suspension system, the Co-TiO$_x$ membrane would effectively enrich these $^1O_2$, $SO_4^{•-}$ and $^•OH$ radicals within its numerous 4.6 Å interlayer free spacings, and thus significantly enhancing the angstrom-confined catalytic performance.

DFT calculations were conducted to calculate the S–O bond length and free energy of PMS in the angstrom-confinement Co-TiO$_x$ membrane. When compared to PMS in vacuum state ($d_{\text{S–O}} = 1.905$ Å), the length of S–O bond in PMS increased with the decreasing interlayer free spacing (Fig. 4c). For hydrated-state Co-TiO$_x$ membrane that contained numerous angstrom-confined channels of 4.6 Å, PMS dissociation occurred through the cleavage of S–O bond with the subsequent generation of $^1O_2$. However, that was not the case at the interlayer free spacings of 8 and 50 Å, where the S–O bond was stretched but not cleaved. As such, theoretically, there will be more $^1O_2$ generated in the 4.6 Å interlayer free spacings of Co-TiO$_x$ membrane compared to the cases with 8 and 50 Å interlayer free spacings, contributing to the enhanced catalytic performance.

The changes in the electron density distribution between the PMS and Co-TiO$_x$ nanosheets were then studied using electron density difference (EDD) analysis (Fig. S14). The large redistributions of electronic densities from S atom of $HSO_3^-$ to O atom of Co-TiO$_x$ nanosheets are observed (S–>O), indicating that PMS adsorb onto the O sites of Co-TiO$_x$ nanosheets. Notably, the electronic redistributions from O atom to Co atom in Co-TiO$_x$ nanosheets are also obtained (O–>Co), suggesting that the substituted Co atoms in Co-TiO$_x$ nanosheets could effectively facilitate the PMS activation. However, much weaker electrostatic interactions were found at the larger interlayer free spacings of 8 and 50 Å compared to the case with 4.6 Å interlayer free spacings (Fig. 4c). The adsorption energy ($E_{\text{ads}}$) of PMS increased from $-0.82$ to $-3.96$ eV when the free interlayer spacing decreased from 50 to 4.6 Å. Hence, the angstrom-confined spaces within the Co-TiO$_x$ membrane can significantly enhance the $E_{\text{ads}}$ of PMS and the radical yields, facilitating the angstrom-confined catalytic performance.

To further elucidate the mechanism of PMS dissociation in the Co-TiO$_x$ membrane, Climbing Image-Nudged Elastic Band (CINEB) calculations were performed. It is noteworthy that the cleavage of S–O bond in the Co-TiO$_x$ membrane with an interlayer free spacing of 4.6 Å occurs, thus a larger interlayer free spacing of 7 Å was used for the simulated PMS dissociation process. As shown in Fig. 4d, a transition state ($+0.09$ eV) with a longer distance between S and O (2.365 Å vs. 1.903 Å) and a relatively unchanged O–O bond length (1.318 Å vs. 1.320 Å) as compared to the initial state were found. This result indicates that the dissociation of PMS is attributed to the cleavage of S–O bond. Moreover, the free energy of $-1.41$ eV indicates that the dissociation of the adsorbed PMS molecules in the angstrom-confined Co-TiO$_x$ membrane is spontaneous and thermodynamically favorable.

In addition, the angstrom-scaled interlayer spacing within Co-TiO$_x$ membrane can greatly enhance the interaction between the ROS and target pollutants with more effective diffusion and reactions, which can also significantly accelerate the catalytic process[2,10]. This is indeed the case as confirmed by the AIMD and MD results. AIMD and MD simulations were performed to further

elucidate the diffusion process of PMS, ranitidine, and water in the 4.6 Å interlayer nanochannels of Co-TiO$_x$ membrane.

The diffusion process of PMS, ranitidine, and water within membrane nanochannels can be divided into two processes, diffusion from aqueous solution into membrane nanochannels and diffusion within membrane nanochannels. According to DFT results (Fig. S8), ranitidine, PMS, and water can enter the 4.6 Å interlayer nanochannels of Co-TiO$_x$ membrane. The larger size ranitidine molecules are more prone to deform, rotate, bend and twist when transporting into the 4.6 Å interlayer nanochannels of Co-TiO$_x$ membrane[27]. The molecular sizes of PMS, ranitidine and water within membrane nanochannels are 1.078, 2.659 and 0.627 Å, respectively (Fig. S8). This is also confirmed by the AIMD results that ranitidine, PMS, and water can enter the 4.6 Å Co-TiO$_x$ membrane nanochannels as shown in Fig. S15a, c. In addition, the diffusion of PMS and ranitidine to the surface of Co-TiO$_x$ nanosheets can be much easier with the greatly shortened migration distance as evidenced by the Mean Square Displacement (MSD) curves as shown in Fig. S16a. As such, it can be expected that the concentration of PMS and ranitidine molecules on the surface of Co-TiO$_x$ nanosheets would be much higher at smaller interlayer-free spacings especially at angstrom-scales (Fig. S17), which could enhance the effective collision of ROS and target pollutants, maximize the utilization of reactive radicals and thus significantly promote the catalytic degradation reactions. This is indeed the case as evidenced by the AIMD results that only after 1500 fs, the PMS and ranitidine molecules decompose into various fragments in the 4.6 Å interlayer nanochannels of Co-TiO$_x$ membrane (Fig. S15, d). As such, the diffusion process of PMS fragments, ranitidine fragments, and water molecules inside the 4.6 Å Co-TiO$_x$ membrane nanochannels was further investigated. As shown in Fig. S16b, the diffusion process of PMS fragments, ranitidine fragments, and water molecules follows the order: water molecules > ranitidine fragments > PMS fragments. MD simulations further show that water molecules as the form of monolayer water molecules rapidly move through the 4.6 Å Co-TiO$_x$ interlayer nanochannels by the driving force of hydrogen bonding (Fig. S18). Water molecules with a highly ordered structure can significantly accelerate the diffusion process at angstrom-scales[17,18] and thus facilitate the catalytic reactions. Moreover, the hydrogen bonding interaction between water molecules and Co-TiO$_x$ nanosheets becomes much stronger at smaller interlayer spacings especially at 4.6 Å (Fig. S19), which could also effectively facilitate the diffusion process within the angstrom-scale nanochannels and thus fast the catalytic reactions[2,10].

Motivated by the collective investigations and analyses, the mechanism of Co-TiO$_x$ membrane-based angstrom-confined catalysis activated with PMS towards ranitidine removal can be described as follows. First, ranitidine and PMS adsorb onto the Co-TiO$_x$ nanosheets during the transport in the 4.6 Å interlayer nanochannels of Co-TiO$_x$ membrane (Fig. S8). Secondly, the adsorbed PMS decomposes into $HSO_3^-$ and $^1O_2$ species with the cleavage of S–O bond in the angstrom-confined spaces within Co-TiO$_x$ membrane (Fig. 4c) (SI, Equation 6). Then, $\equiv$Co(II) serve as the reactive sites for the reduction of $HSO_5^-$ to $SO_4^{•-}$ via a one-electron process with the transformation of $\equiv$Co(II) to $\equiv$Co(III) (SI, Equation 7), and $SO_4^{•-}$ is further hydrolyzed to $^•OH$ (Fig. 4b and S13) (SI, Equation 8). The concomitant reduction of $\equiv$Co(III) to $\equiv$Co(II) occurs when $HSO_5^-$ is oxidized to $SO_5^{•-}$ by $\equiv$Co(III) (SI, Equation 9). Therefore, the conversion and regeneration of $\equiv$Co(II)/$\equiv$Co(III) cycle (Fig. S5 and Table S3) can provide a continuous generation of ROS, which can then support a long-term and efficient degradation of organic pollutants (SI, Equation 10).

The degradation products of ranitidine and their toxicity after the Co-TiO$_x$ membrane-based angstrom-confined catalysis were further

investigated. Twelve intermediates (P1-P12) generated during the ranitidine degradation process were identified as shown in Fig. S20 and Table S5. The Ecological Structure-Activity Relationship (ECOSAR) class program (ECOSAR v2.2, EPA) was utilized to predict the acute toxicity ($LC_{50}$ or $EC_{50}$) and chronic toxicity (ChV) of ranitidine and its intermediates to three aquatic organisms, including fish, daphnid, and algae. As shown in Fig. S21, most of the degradation products except for P8 are not harmful ($LC_{50}/EC_{50}/ChV > 100$ mg/L), and their toxicity are much lower than ranitidine. Moreover, after 20 min filtration reaction, the degradation products were reduced drastically. P8 disappeared, and P11 (not harmful product) dominated the final degradation products. As such, it can be concluded that the Co-TiO$_x$ membrane-based angstrom-confined catalysis not only efficiently removed water contaminants but also significantly decreased their toxicity.

The confinement effects of other membranes assembled with alternative 2D materials, e.g., graphene and boron nitride (BN) with vacancy defects, molybdenum disulfide (MoS$_2$), and black phosphorus (P), were also investigated with DFT calculations (Fig. 4e). Clearly, a significant increase in the adsorption energy ($E_{ads}$) of PMS within the angstrom-confined space was observed for 2D membranes with the interlayer free spacings of below ~10 Å. This is then followed by the thermodynamically favorable spontaneous dissociation of PMS especially at an interlayer free spacing of 5 Å. Therefore, based on these DFT calculations, the AOP performance can be significantly improved by adopting a rational design of the angstrom-confined catalytic membranes, and such a strategy can be also extended to other 2D material-assembled membranes. Hence, this work unraveled the importance of utilizing angstrom-confinement strategy in the design of efficient catalysts for water purification. Our work can serve as a design blueprint for the development of other angstrom-confinement catalysis systems, and also provide breakthroughs in AOPs research.

## Methods

**Synthesis of Co-TiO$_x$ nanosheets and Co-TiO$_x$ membranes**. The preparation process includes four stages, namely, heating, protonation, expansion, and solution-based exfoliation. In stage I, a stoichiometric mixture of TiO$_2$ (0.25 mol, 20 g), CoO (0.06 mol, 4.69 g), and K$_2$CO$_3$ (0.06 mol, 5.94 g) was annealed two times at 1000 °C with an interval grinding treatment to fabricate mixed alkali metal titanate of K$_{0.8}$Ti$_{(5.2-y)/3}$Co$_y$O$_4$ (y = 0 for 2D TiO$_x$ and y = 0.4 for 2D Co-TiO$_x$). In stage II, the mixed alkali metal titanate (1 g) was stirred in 200 ml of HCl (1 M) solution for 4 days to achieve sufficient ion exchange of K$^+$ ions with H$^+$. The resulting mixture was washed with deionized water to remove acid residues and dried in oven at 80 °C for 12 h to obtain the protonated sediment H$_{(3.2-2y)/}$ $_3$Ti$_{(5.2-y)/3}$Co$_y$O$_4$. In stage III, the protonic titanate was soaked in ~10 % (w/v) TBAOH aqueous solution (H$^+$:TBA$^+$ = 1:1 in molar ratio) for 5 h to generate expanded titanate of TBA$_z$H$_{(3.2-2y)/3-z}$Ti$_{(5.2-y)/3}$Co$_y$O$_4$. In stage IV, the expanded titanate was exfoliated into unilamellar titanate platelets in an appropriate amount of deionized water by mechanical shaking for two days to obtain a stable suspension of 2D TiO$_x$ and Co-TiO$_x$. The Co-TiO$_x$ membranes were fabricated by vacuum filtration of the as-prepared Co-TiO$_x$ nanosheets dispersion (5 μg/L, 200 mL) with a MCE membrane as the substrate and dried in a vacuum desiccator overnight. Co-TiO$_x$ membranes with different Co-TiO$_x$ loadings (i.e., 0.012, 0.016, 0.024, 0.04, 0.08, 0.12, 0.16 mg/cm$^2$) were prepared to investigate their effect on permeate flux and pollutant removal.

**Characterization**. X-ray diffraction (XRD) (λ = 0.15418 nm, Bruker D8 Advance, Germany) was used to analyze the phase structure of Co-TiO$_x$ nanosheets and d-spacing of membranes. The morphology and thickness of Co-TiO$_x$ nanosheets were identified by atomic force microscopy (AFM) (Cyper ES, Oxford Instruments, UK). Morphology, lattice structure, and the interlayer spacing of Co-TiO$_x$ membrane were observed using high-resolution transmission electron microscopy (HRTEM, JEOL 2100F, Japan). The surface and cross-section morphologies of Co-TiO$_x$ membrane were observed using scanning electron microscopy (SEM) (Hitachi SU8010, Japan) with energy dispersive X-ray spectroscopy (EDX) analysis. The atomic composition and chemical structures of Co-TiO$_x$ membrane were confirmed by X-ray photoelectron spectroscopic (XPS) (PHI5000 VersaProbeII, Japan) with a monochromated Al Kα radiation at 1486.6 eV. The concentration of ranitidine was detected using liquid chromatography/tandem mass spectrometry (LC-MS, 8050, Shimadzu, Japan). Electron paramagnetic resonance (EPR) spectra were obtained using MS-5000 spectrometer (Bruker, Germany). The Ecological Structure-Activity Relationship (ECOSAR) class program (ECOSAR v2.2, EPA) was utilized to

predict the acute toxicity ($LC_{50}$ or $EC_{50}$) and chronic toxicity (ChV) of ranitidine and its intermediates to three aquatic organisms, including fish, daphnid, and algae.

**Catalytic activity measurements**. The catalytic degradation experiments were carried out on a dead-end filtration unit under an operating pressure of 0.1 MPa at room temperature. For the angstrom-confined catalysis experiment, the mixed solution of ranitidine (5 mg/L) and PMS (0.16 mM) was filtrated through Co-TiO$_x$ membrane, and then the permeate was detected by LC-MS (the detailed operation method is shown in Table S4). As the concentration of permeate after 4.5 min was under the detection limit of ranitidine (0.1 μg/L), the removal efficiency of ranitidine is calculated as ~100%. The flux of membranes J (L m$^{-2}$ h$^{-1}$) was calculated using the following equation:

$$J = \frac{q}{S \times t} \qquad (1)$$

where q (L) is the volume of permeation solution, S (m$^2$) represents the effective filtration area of membranes, and t (h) is experimental time. The removal efficiency R (%) was calculated using the following equation:

$$R = \frac{C_F - C_P}{C_F} \times 100\% \qquad (2)$$

where $C_P$ (ppm) and $C_F$ (ppm) are the ranitidine concentrations in the permeate and feed, respectively.

For batch suspension reaction, 1 mg Co-TiO$_x$ nanosheets were added into 60 mL ranitidine solution (5 mg/L), and the suspension was stirred for 15 min to achieve an adsorption-desorption equilibrium. PMS (0.16 mM) was then added into the suspension to initiate the reaction, and the mixture was continually stirred at a speed of 300 rpm. The samples containing 1 mL of the above solution were taken at given time intervals during the reaction process and filtered by a cellulose acetate membrane (0.02 μm pore size). The filtrates were collected separately to determine the removal efficiency using LC-MS.

**Quenching experiment**. The tert-butanol (TBA, 90 mM), Methanol (90 mM), p-benzoquinone (p-BQ, 1 mM), and L-histidine (L-HIS, 12 mM) were employed to quench hydroxyl radicals, sulfate radicals, superoxide ion radicals, and singlet oxygen, respectively. Typically, scavengers were added into the mixed solution of ranitidine (5 mg/L) and PMS (0.16 mM) separately, and then the solution was filtrated through Co-TiO$_x$ membrane. The permeate was detected using LC-MS to quantify the contribution of various radicals.

**Computational methodology**

*DFT*. The periodical density functional theory (DFT) calculations were employed by the Vienna Ab initio Simulation Package (VASP) package[28–31]. The Perdew-Burke-Ernzerhof (PBE) exchange-correlation functions[32], with a projector-augmented wave (PAW) scheme[33], were applied to describe the ion-electron exchange-correlation. The DFT-D3[34] including Becke-Johnson damping[35] was utilized to correct the long-range dispersion. The energy cutoff for the plane-wave expansion was setup to 450 eV. The Brillouin zone integration was generated according to the Monkhorst-Pack method[36] using a Gamma centered $2 \times 2 \times 1$ k-point mesh. All structures were relaxed until the energy changed less than $1 \times 10^{-5}$ eV, with the forces on each atom below 0.01 eV/Å.

To model the Co-doping lepidocrocite TiO$_2$ nanosheets, a supercell of 36 Ti atoms and 72 O atoms was constructed and optimized, and then 7 Co atoms were used to substitute the Ti atoms to simulate the practical elemental composition (Fig. 2e). The adsorption energy ($E_{ads}$) was calculated as

$$E_{ads} = E_{system} - E_{surface} - E_{molecule} \qquad (3)$$

where the $E_{system}$ is the energy of the optimized system; $E_{surface}$ is the energy of the bare surface; $E_{molecule}$ is the energy of an optimized PMS/ranitidine/H$_2$O molecule within a $30 \times 30 \times 30$ box.

To illustrate the PMS dissociation process, the transition states were located utilizing the Climbing Nudged Elastic Band (CINEB) method implemented in VASP-VTST code[37]. After the CINEB calculations, the frequency calculations were performed with a numerical algorithm and atomic displacement of 0.015 Å. A true transition state from CINEB calculations was confirmed by a single negative frequency. The free energy corrections were accomplished with VASPKIT program[38] at a temperature of 298.15 K.

Ab initio molecular dynamics (AIMD) simulations were performed to investigate the diffusion process of PMS, ranitidine, and water within the 4.6 Å interlayer free spacing of Co-TiO$_x$ nanosheets. A cell of 27.261 Å × 22.398 Å × 8.844 Å was constructed, containing 1 ranitidine, 4 PMS, and 10 H$_2$O molecules. A nose thermostat with 0.5 fs time step was applied.

*MD*. Molecular Dynamic (MD) simulations using the Forcite module in Material Studio package were performed to illustrate the transport properties of PMS, ranitidine, and water within the interlayer of Co-TiO$_x$ nanosheets[39,40]. Two modules including solvent module and crystal (Co-TiO$_x$) module were considered. The solvent module having 1 ranitidine molecule, 5 PMS molecules and 200 water molecules was built by AC module through Drieding force field. The configuration

of the crystal model was calculated by DFT and the crystal module was transformed into one slab, Co-TiO$_x$ (0 1 0). Then all modules were set at 298.15 K with 0.1 MPa to run the MD process. Note that the MD process with the solvent module having 1 ranitidine molecule, 5 PMS molecules and 200 water molecules requires the minimal interlayer free spacing of Co-TiO$_x$ of 2.5 nm. As such, three different interlayer-free spacings (2.5, 3.5, 5 nm) of Co-TiO$_x$ were investigated by MD to compare the mass transfer performance. Geometry optimization was performed first to have the stable state of the molecules followed by the 20 ps-NVT and 20 ps-NPT processes to verify the most realistic structures. Finally, the NVT process was performed to present the entire transport process[41,42]. The time of the final NVT process for the interlayer free spacings of 2.5 and 3.5 nm is 80 ps, while that is 100 ps for the interlayer free spacing of 5 nm because of its larger calculation environment.

Universal force field was selected to represent the interaction between microscopic atoms. We chose a 0.7-time step to ensure the stability of the bonded interactions. During all the MD processes, we exported 1 frame every 250 steps to collect efficient data for the Mean Square Displacement (MSD) curve. The Ewald method was selected for the correction of long-range electrostatic interactions. The electrostatic and van der Waals summation methods were set as Ewald and Atom based methods, respectively. Cut off distance was 12.5 Å with a 0.5 Å buffer width.

## Data availability

All data are available in the main text or Supplementary Information.

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

## Acknowledgements

This work was financially supported by the National Natural Science Foundation of China (52170041), Tsinghua SIGS Start-up Funding (QD2020002N), the Committee of Science and Technology Innovation of Shenzhen (JCYJ20190813163401660), and China Postdoctoral Science Foundation (2020M682893).

## Author contributions

Z.Z. conceived the project. B.D. and Z.H. synthesized the Co-TiO$_x$ nanosheets and assisted in material characterization. C.M. fabricated the Co-TiO$_x$ membrane and performed the catalytic activity experiments. C.M. and Z.Z. performed the membrane characterization. S.Z. performed the theoretical calculations. L.C. performed the toxicity analysis. C.M. and Z.Z. analysed the results and wrote the manuscript. K.O., B.Y. and J.K. commented on the manuscript. All authors discussed, commented on, and revised the manuscript.

## Competing interests

The authors declare no competing interests.
