## [Peer Review File · Nature Communications]

Title: Angstrom-confined catalytic water purification within Co-TiOx laminar membrane nanochannelsREVIEWER COMMENTS

Reviewer #1 (Remarks to the Author):

Ref the Manuscript "Angstrom-confined catalytic water purification within Co-TiO_x laminar membrane nanochannels" by Zhang et al.

The manuscript proposes a new laminar membrane that has been exhaustively characterized using high resolution techniques associated to DFT calculations. However, the most important part of the manuscript is that related to its catalytic behavior and to the correlation of all the collected characteristics with the catalytic behavior. Unfortunately, the system did not well in this direction. When we discuss on water removal, an efficiency smaller than 10% is irrelevant. Also important, in this case the conversion is not the main indicator but the selectivity because during the reaction you may produce even more toxic molecule than the initial one. The manuscript does not analyze this aspect. Based on this I can not recommend its publication in the Nature Communication journal.

Reviewer #2 (Remarks to the Author):

Comments on the manuscript "Angstrom confined catalytic water purification within Co TiO_x laminar membrane nanochannels" by Chenchen Meng et al.

The manuscript contains many interesting results. The preparation and characterization techniques of the Co TiO_x membrane are extensively described. In general, the experimental results are well explained. I recommend accepting this manuscript.

Very Minor Revisions

Manuscript

Line 151: the symbol is not clear.

Lines 230-232: The two compared quantities should have the same unit.

Line 232: The sentence "Such a rate is also 5 7 orders of magnitude faster than that achieved by other means" is not confirmed by the values reported in Table S1!

Supplementary Information

Line 53: Expanded

Line 181: Calculation?

Table S1: The last line should be checked.

Table S2: kW, plasma

Table S3: Before, E(eV),

Table S3: What is the meaning of the quantity A?

Reviewer #3 (Remarks to the Author):

In this work Zhang et al reported a very interesting and important work on the development of angstrom-level confined catalysis system for PMS activation and organic pollutant degradation. The proposed Co-TiO_x laminar membrane nanochannels coupled with PMS activation exhibited orders of magnitude faster degradation kinetics toward various pollutants than other heterogeneous analogues, and the underlying mechanism for the unique process has been systematically probed. In general, I believe this work will help to better understand nanoconfined catalysis, whether in terms of method development or mechanism explanation. Thus, I am sure that this work deserves publication in NC after considering the following comments: (1) The authors have made considerable efforts on the mechanism probing such as the formation of active species. I just wonder if there are any new phenomena on the active species for the confined system over the reference one? Three key species were identified to be responsible for the organic degradation, and any difference from the reference system? For instance, the percentage for each species? On the other side, what is the difference of the oxidation products of both systems? These new evidence, if available, may be useful to better understand such confined catalysis; (2) As for the extremely fast kinetics of the new system, the authors are advised to propose a reasonable approach for such new findings. Why faster to such a high level? What is the contribution of the diffusion of the reactants inside the channel and how to evaluate such process? DFT and other used models may not provide very solid evidence for such unique performance. (3) some minor comments: L 157, 3.67? and 3.76? typo for further check; Fig. 3 D, it seems unnecessary for such comparison in scientific viewpoint.

Manuscript ID: NCOMMS-22-07845

**Angstrom-confined catalytic water purification within Co-TiO_x laminar membrane
nanochannels**

REVIEWER 1

General Comment: The manuscript proposes a new laminar membrane that has been exhaustively characterized using high resolution techniques associated to DFT calculations. However, the most important part of the manuscript is that related to its catalytic behavior and to the correlation of all the collected characteristics with the catalytic behavior. Unfortunately, the system did not well in this direction.

Response: Thanks for your suggestions on this manuscript. We have significantly strengthened the mechanism part by rephrasing and adding more discussions. Meanwhile, we have also supplemented molecular dynamics (MD) simulations to illustrate the transport properties of PMS, ranitidine, and water within the interlayer of Co-TiO_x nanosheets. Overall, towards understanding the angstrom-confinement catalysis, we consider three perspectives, enrichment of reactants, more effective diffusion and reactions, and stronger electronic interactions at angstrom-scales, which have been theoretically-proposed to be the potential main contributors to the enhanced angstrom-confined catalytic performance. Herein, the mechanism of the Co-TiO_x membrane-based angstrom-confined catalysis activated with PMS was investigated using reactive oxygen species (ROS) scavenging experiments, electron paramagnetic resonance (EPR), DFT and MD. These have been supplemented in the revised manuscript as shown below.

“Towards understanding the angstrom-confinement catalysis, enrichment of reactants, more effective diffusion and reactions, and stronger electronic interactions at angstrom-scales have been theoretically-proposed to be the potential main contributors to the enhanced catalytic performance^{2,10}. As such, the mechanism of the Co-TiO_x membrane-based angstrom-confined catalysis activated with PMS was investigated using ROS scavenging experiments, electron paramagnetic resonance (EPR), DFT and molecular dynamics (MD).” (Page 15, Line 280–286)

“..... Compared with the batch suspension system, the Co-TiO_x membrane would effectively enrich these ¹O₂, SO₄^{•-} and •OH radicals within its numerous 4.6 Å interlayer free spacings, and thus significantly enhancing the angstrom-confined catalytic performance.” (Page 16, Line 318–321)

“..... For hydrated-state Co-TiO_x membrane that contained numerous angstrom-confined channels of 4.6 Å, PMS dissociation occurred through the cleavage of S–O bond with the subsequent generation of ¹O₂. However, that was not the case at the interlayer free spacings of 8 and 50 Å, where the S–O bond was stretched but not cleaved. As such, theoretically, there will be more ¹O₂ generated in the 4.6 Å interlayer free spacings of Co-TiO_x membrane compared to the cases with 8 and 50 Å interlayer free spacings, contributing to the enhanced catalytic performance.” (Page 17, Line 329–333)

“.....However, much weaker electrostatic interactions were found at the larger interlayer free spacings of 8 and 50 Å compared to the case with 4.6 Å interlayer free spacings (Fig. 4C). The adsorption energy (*E*_{ads}) of PMS increased from -0.82 to -3.96 eV when the free interlayer

spacing decreased from 50 to 4.6 Å. Hence, the angstrom-confined spaces within the Co-TiO_x membrane can significantly enhance the E_{ads} of PMS and the radical yields, facilitating the angstrom-confined catalytic performance.” (Page 18, Line 342–347)

“In addition, the angstrom-scaled interlayer spacing within Co-TiO_x membrane can greatly enhance the interaction between the ROS and target pollutants with more effective diffusion and reactions, which can also significantly accelerate the catalytic process^{2,10}. This is indeed the case as confirmed by the MD results. MD simulations were performed to illustrate the transport properties of PMS, ranitidine, and water within the interlayer of Co-TiO_x nanosheets. Note that the MD process with the solvent module having 1 ranitidine molecule, 5 PMS molecules and 200 water molecules requires the minimal interlayer free spacing of Co-TiO_x of 2.5 nm. As such, three different interlayer-free spacings (2.5, 3.5, 5 nm) of Co-TiO_x were investigated by MD to compare the mass transfer performance. As shown in Fig. S15, the diffusion of PMS, ranitidine, and H₂O molecules to the surface of Co-TiO_x nanosheets can be much easier with the greatly shortened migration distance as evidenced by the Mean Square Displacement (MSD) curve with the decreasing interlayer-free spacing. As such, it can be expected that the concentration of PMS and ranitidine molecules on the surface of Co-TiO_x nanosheets would be much higher at smaller interlayer-free spacings especially at angstrom-scales, which could enhance the effective collision of ROS and target pollutants, maximize the utilization of reactive radicals and thus significantly promote the catalytic degradation reactions. Moreover, the hydrogen bonding interaction between water molecules and Co-TiO_x nanosheets becomes much stronger at smaller interlayer spacings especially at 4.6 Å (Fig. S16), which could also effectively facilitate the diffusion process within the angstrom-scale nanochannels and thus fast the catalytic

reactions^{2,10}.” (Page 18-19, Line 361–382)

Comment 1: When we discuss on water removal, an efficiency smaller than 10% is irrelevant.

Response: In this work, Co-TiO_x membrane/PMS system exhibited ~100% effective removal for diverse water contaminants with a retention time of < 30 ms (Fig. 3B-F). As a control experiment, in the absence of PMS, Co-TiO_x membrane exhibited very poor ranitidine removal performance (< 20%) merely relied on the molecular sieving and adsorption mechanism (Fig. 3B), and the removal efficiency decreased over time and eventually ceased after 30 min due to the adsorption saturation of the membrane. As such, it can be concluded that ranitidine removal occurred was mainly *via* catalytic degradation. This has been indicated in the manuscript and also shown below.

“In the absence of PMS, both Co-TiO_x and TiO_x membranes exhibited also very poor ranitidine removal performances (< 20%) as they merely relied on the molecular sieving and adsorption mechanism. As such, the removal efficiency decreased over time, and further ranitidine removal eventually ceased after 30 min due to the adsorption saturation in the membrane. In addition, TiO_x membrane without Co dopants also exhibited a low ranitidine removal efficiency of 20.3% within 30 min. This result suggests the importance of Co ion sites in Co-TiO_x lattice for PMS activation and catalytic reactions. Consistently, a suspension of Co-TiO_x nanosheets and PMS achieved a much higher ranitidine removal efficiency of 85.4% within 20 min.” (Page 11, Line 217–226)

“In marked contrast, the ACC enabled by the Co-TiO_x layered membrane produced the extremely short calculated retention time of only 5.5 ms (Fig. S9) for the 100% ranitidine removal (Fig. 3C). Moreover, the first-order rate constant of the ACC ranitidine degradation process was 63600

min⁻¹ (inset Fig. 3C), which is about 6 orders of magnitude faster than that for the non-confined heterogeneous Co-TiO_x nanosheets suspension (0.092 min⁻¹) (Fig. S10). Such a rate is also 5–7 orders of magnitude faster than that achieved by other means (0.0032-0.33 min⁻¹) (Fig. 3D and Table S1). Indeed, these results imply that designing the Co-TiO_x membrane with the angstrom-scale spatial confinement can significantly enhance the performance compared to common catalytic systems. The achieved high catalytic activity towards ranitidine can be attributed to the angstrom-confinement effect of the Co-TiO_x membrane.” (Page 12, Line 228–239)

Comment 2: Also important, in this case the conversion is not the main indicator but the selectivity because during the reaction you may produce even more toxic molecule than the initial one.

Response: We appreciate your comment. The main purpose of this study is to design atomistically and demonstrate experimentally for the first time about the generic angstrom-confined catalytic water contaminant degradation concept. In order to address the reviewer’s concern, we have supplemented the degradation products of ranitidine and their toxicity in Figs. S17 and S18, and Table S5. Twelve intermediates (P1-P12) generated during the ranitidine degradation process were identified as shown in Fig. S17 and Table S5. The Ecological Structure-Activity Relationship (ECOSAR) class program (ECOSAR v2.2, EPA) was utilized to predict the acute toxicity (LC₅₀ or EC₅₀) and chronic toxicity (ChV) of ranitidine and its intermediates to three aquatic organisms, including fish, daphnid, and algae. As shown in Fig. S18, most of the degradation products except for P8 are not harmful (LC₅₀/EC₅₀/ChV > 100 mg/L), and their toxicity are much lower than ranitidine. Moreover, after 20 min filtration reaction, the degradation products were reduced drastically. P8 disappeared, and P11 (not

harmful product) dominated the final degradation products. As such, it can be concluded that the Co-TiO_x membrane-based angstrom-confined catalysis not only efficiently removed water contaminants but also significantly decreased their toxicity. The relevant descriptions have been supplemented in the revised manuscript as shown below.

“The degradation products of ranitidine and their toxicity after the Co-TiO_x membrane-based angstrom-confined catalysis were further investigated. Twelve intermediates (P1-P12) generated during the ranitidine degradation process were identified as shown in Fig. S17 and Table S5. The Ecological Structure-Activity Relationship (ECOSAR) class program (ECOSAR v2.2, EPA) was utilized to predict the acute toxicity (LC₅₀ or EC₅₀) and chronic toxicity (ChV) of ranitidine and its intermediates to three aquatic organisms, including fish, daphnid, and algae. As shown in Fig. S18, most of the degradation products except for P8 are not harmful (LC₅₀/EC₅₀/ChV > 100 mg/L), and their toxicity are much lower than ranitidine. Moreover, after 20 min filtration reaction, the degradation products were reduced drastically. P8 disappeared, and P11 (not harmful product) dominated the final degradation products. As such, it can be concluded that the Co-TiO_x membrane-based angstrom-confined catalysis not only efficiently removed water contaminants but also significantly decreased their toxicity.” (Page 20, Line 399–411)

Fig. S17. Spectra of ranitidine degradation products obtained in ESI(+)-MS mode by LC-MS. (A) feed solution ($t = 0$ min); permeate solutions (B) ($t = 2$ min) and (C) ($t = 20$ min).

Table S5. Degradation products of ranitidine in the Co-TiO_x membrane/PMS system.

Compounds	m/z value	Structure
Ranitidine	315	<chem>CN(C)Cc1cc(OCCN(C)C)oc1</chem>
P1	369	<chem>CN(C)Cc1cc(OCC(=O)N(C)C)oc1</chem>
P2	339	<chem>CN(C)Cc1cc(OCC(=O)N)oc1</chem>
P3	323	<chem>CN(C)Cc1cc(OCC(=O)O)oc1</chem>
P4	309	<chem>CN(C)Cc1cc(OCC(=O)O)oc1</chem>
P5	293	<chem>CN(C)Cc1cc(OCC(=O)O)oc1</chem>

P6	279	P7	263	P8	242	P9	156	P10	105	P11	80	P12	64	

Fig. S18. Toxicity estimation of ranitidine and its degradation intermediates using the ECOSAR program. According to the system established by the Globally Harmonized System of Classification and Labeling of Chemicals (GHS), the predicted toxicity values of ranitidine and all intermediates can be divided into four categories: very toxic ($LC_{50}/EC_{50}/ChV < 1$ mg/L), toxic (1 mg/L $< LC_{50}/EC_{50}/ChV < 10$ mg/L), harmful (10 mg/L $< LC_{50}/EC_{50}/ChV < 100$ mg/L), and not harmful ($LC_{50}/EC_{50}/ChV > 100$ mg/L).

REVIEWER 2

General Comment: The manuscript contains many interesting results. The preparation and characterization techniques of the Co-TiO_x membrane are extensively described. In general, the experimental results are well explained. I recommend accepting this manuscript.

Response: Thank you for your recognition of this work and your suggestions on this manuscript.

A point-by-point response on all issues raised is provided below.

Comment 1: Line 151: the symbol is not clear.

Response: This has been addressed as shown below. Thanks.

“...there was an increase in the *d*-spacing from 3.67 Å to 3.76 Å...”

Comment 2: Lines 230–232: The two compared quantities should have the same unit.

Response: Thank you for the suggestion. We have unified the units in the revised manuscript and changed 1.06 ms⁻¹ to 63600 min⁻¹.

Comment 3: The sentence “Such a rate is also 5–7 orders of magnitude faster than that achieved by other means” is not confirmed by the values reported in Table S1.

Response: The first-order rate constants of the ranitidine degradation process by Co-TiO_x membrane/PMS system and other recently reported technologies are shown in Table S1. The *k* value for the Co-TiO_x membrane/PMS system is 63600 min⁻¹, which is 5–7 orders of magnitude higher than those of the conventional heterogeneous catalytic systems (0.0032–0.33 min⁻¹, Table S1). We have unified the unit and changed 1.06 ms⁻¹ to 63600 min⁻¹ in Table S1 and supplemented the rate constant of 63600 min⁻¹ in the sentence as shown below.

“Such a rate (63600 min⁻¹) is also 5–7 orders of magnitude faster than that achieved by other means (0.0032–0.33 min⁻¹)” (Page 12, Line 233–234)

Comment 4: Line 53: Expanded

Response: Thanks. We have corrected the typo in the revised Supplementary Materials (Page 2, Line 54)

Comment 5: Line 181: Calculation?

Response: Thanks. We have corrected the typo in the revised Supplementary Materials (Page 13, Line 212)

Comment 6: Table S1: The last line should be checked.

Response: Thank you for the suggestion. The first-order rate constant of the ACC ranitidine degradation process is 1.06 ms^{-1} (inset Fig. 3C, Page 14). Here, in order to better compare with other reported values, we have unified the unit and changed 1.06 ms^{-1} to 63600 min^{-1} in Table S1. Co-TiO_x membrane with a thickness of 500 nm was selected since the 100% removal efficiency and permeance of $131 \text{ L}\cdot\text{m}^{-2}\cdot\text{h}^{-1}\cdot\text{bar}^{-1}$ were considered to be the optimal ACC conditions. The membrane thickness of ~500 nm corresponds to the Co-TiO_x loading of 1 mg ($0.08 \text{ mg}/\text{cm}^2$). We have indicated the loading amount of $0.08 \text{ mg}/\text{cm}^2$ in Table S1 and supplemented the related description in the revised manuscript as shown below.

“Therefore, Co-TiO_x membrane with a thickness of 500 nm, corresponding to the Co-TiO_x loading of $0.08 \text{ mg}/\text{cm}^2$, was selected since the 100% removal efficiency and permeance of $131 \text{ L}\cdot\text{m}^{-2}\cdot\text{h}^{-1}\cdot\text{bar}^{-1}$ were considered to be the optimal ACC conditions.” (Page 10, Line 202–205)

“Co-TiO_x membranes with different Co-TiO_x loadings (i.e., 0.012, 0.016, 0.024, 0.04, 0.08, 0.12, $0.16 \text{ mg}/\text{cm}^2$) were prepared to investigate their effect on permeate flux and pollutant removal.” (Supplementary Materials, Page 3, Line 60–62)

Comment 7: Table S2: kW, plasma

Response: Thanks. We have corrected the typo in Table S2 in the revised Supplementary Materials (Page 24, Line 319)

Comment 8: Table S3: Before, E(eV),

Response: Thanks. We have corrected the typo in Table S3 in the revised Supplementary Materials (Page 24, Line 321)

Comment 9: What is the meaning of the quantity A?

Response: A represents the peak area and it has been changed to “Peak area” in Table S3.

REVIEWER 3

General Comment: In this work Zhang et al reported a very interesting and important work on the development of angstrom-level confined catalysis system for PMS activation and organic pollutant degradation. The proposed Co-TiO_x laminar membrane nanochannels coupled with PMS activation exhibited orders of magnitude faster degradation kinetics toward various pollutants than other heterogeneous analogues, and the underlying mechanism for the unique process has been systematically probed. In general, I believe this work will help to better understand nanoconfined catalysis, whether in terms of method development or mechanism explanation. Thus, I am sure that this work deserves publication in NC after considering the following comments.

Response: Thank you for your recognition of this work and your suggestions on this manuscript. A point-by-point response on all issues raised is provided below.

Comment 1: The authors have made considerable efforts on the mechanism probing such as the formation of active species. I just wonder if there are any new phenomena on the active species

for the confined system over the reference one? Three key species were identified to be responsible for the organic degradation, and any difference from the reference system? For instance, the percentage for each species? On the other side, what is the difference of the oxidation products of both systems? These new evidences, if available, may be useful to better understand such confined catalysis.

Response: We appreciate your important comments. In this work, Co-TiO_x membrane/PMS system exhibited ~100% effective removal for diverse water contaminates with a retention time of < 30 ms (Fig. 3B-F). However, it is impossible to collect the permeate within 30 ms. Instead, the available permeate for practical measurement is after 4.5 min (the first measured point in Fig. 3B). As such, it is hard to compare the difference between the confined system and the reference one just based on conventional characterizations. However, indeed, we think that there are four new phenomena occurred in the confined system compared to the reference one by the DFT and MD analyses. 1) For hydrated-state Co-TiO_x membrane that contained numerous angstrom-confined channels of 4.6 Å, PMS dissociation occurred through the cleavage of S–O bond with the subsequent generation of ¹O₂. However, that was not the case at the interlayer free spacings of 8 and 50 Å, where the S–O bond was stretched but not cleaved. As such, theoretically, there will be more ¹O₂ generated in the 4.6 Å interlayer free spacings of Co-TiO_x membrane compared to the cases with 8 and 50 Å interlayer free spacings, contributing to the enhanced catalytic performance. 2) Much weaker electrostatic interactions were found at the larger interlayer free spacings of 8 and 50 Å compared to the case with 4.6 Å interlayer free spacings (Fig. 4C). The adsorption energy (E_{ads}) of PMS increased from -0.82 to -3.96 eV when the free interlayer spacing decreased from 50 to 4.6 Å. Hence, the angstrom-confined spaces within the Co-TiO_x membrane can significantly enhance the E_{ads} of PMS and the radical yields, facilitating the

angstrom-confined catalytic performance. **3)** The diffusion of PMS, ranitidine, and H₂O molecules to the surface of Co-TiO_x nanosheets can be much easier with the greatly shortened migration distance as evidenced by the Mean Square Displacement (MSD) curve with the decreasing interlayer-free spacing (Fig. S15). As such, it can be expected that the concentration of PMS and ranitidine molecules on the surface of Co-TiO_x nanosheets would be much higher at smaller interlayer-free spacings especially at angstrom-scales, which could enhance the effective collision of reactive oxygen species (ROS) and target pollutants, maximize the utilization of reactive radicals and thus significantly promote the catalytic degradation reactions. **4)** The hydrogen bonding interaction between water molecules and Co-TiO_x nanosheets becomes much stronger at smaller interlayer spacings especially at 4.6 Å (Fig. S16), which could also effectively facilitate the diffusion process within the angstrom-scale nanochannels and thus fast the catalytic reactions. These have been supplemented in the revised manuscript as shown below.

“..... Compared with the batch suspension system, the Co-TiO_x membrane would effectively enrich these ¹O₂, SO₄⁻ and •OH radicals within its numerous 4.6 Å interlayer free spacings, and thus significantly enhancing the angstrom-confined catalytic performance.” (Page 16, Line 318–321)

“..... For hydrated-state Co-TiO_x membrane that contained numerous angstrom-confined channels of 4.6 Å, PMS dissociation occurred through the cleavage of S–O bond with the subsequent generation of ¹O₂. However, that was not the case at the interlayer free spacings of 8 and 50 Å, where the S–O bond was stretched but not cleaved. As such, theoretically, there will be more ¹O₂ generated in the 4.6 Å interlayer free spacings of Co-TiO_x membrane compared to the

cases with 8 and 50 Å interlayer free spacings, contributing to the enhanced catalytic performance.” (Page 17, Line 329–333)

“.....However, much weaker electrostatic interactions were found at the larger interlayer free spacings of 8 and 50 Å compared to the case with 4.6 Å interlayer free spacings (Fig. 4C). The adsorption energy (E_{ads}) of PMS increased from -0.82 to -3.96 eV when the free interlayer spacing decreased from 50 to 4.6 Å. Hence, the angstrom-confined spaces within the Co-TiO_x membrane can significantly enhance the E_{ads} of PMS and the radical yields, facilitating the angstrom-confined catalytic performance.” (Page 18, Line 342–347)

“In addition, the angstrom-scaled interlayer spacing within Co-TiO_x membrane can greatly enhance the interaction between the ROS and target pollutants with more effective diffusion and reactions, which can also significantly accelerate the catalytic process^{2,10}. This is indeed the case as confirmed by the MD results. MD simulations were performed to illustrate the transport properties of PMS, ranitidine, and water within the interlayer of Co-TiO_x nanosheets. Note that the MD process with the solvent module having 1 ranitidine molecule, 5 PMS molecules and 200 water molecules requires the minimal interlayer free spacing of Co-TiO_x of 2.5 nm. As such, three different interlayer-free spacings (2.5, 3.5, 5 nm) of Co-TiO_x were investigated by MD to compare the mass transfer performance. As shown in Fig. S15, the diffusion of PMS, ranitidine, and H₂O molecules to the surface of Co-TiO_x nanosheets can be much easier with the greatly shortened migration distance as evidenced by the Mean Square Displacement (MSD) curve with the decreasing interlayer-free spacing. As such, it can be expected that the concentration of PMS and ranitidine molecules on the surface of Co-TiO_x nanosheets would be much higher at smaller

interlayer-free spacings especially at angstrom-scales, which could enhance the effective collision of ROS and target pollutants, maximize the utilization of reactive radicals and thus significantly promote the catalytic degradation reactions. Moreover, the hydrogen bonding interaction between water molecules and Co-TiO_x nanosheets becomes much stronger at smaller interlayer spacings especially at 4.6 Å (Fig. S16), which could also effectively facilitate the diffusion process within the angstrom-scale nanochannels and thus fast the catalytic reactions^{2,10}.” (Page 18-19, Line 361–382)

Comment 2: As for the extremely fast kinetics of the new system, the authors are advised to propose a reasonable approach for such new findings. Why faster to such a high level? What is the contribution of the diffusion of the reactants inside the channel and how to evaluate such process? DFT and other used models may not provide very solid evidence for such unique performance.

Response: Thanks for your suggestion. Towards understanding the angstrom-confinement catalysis, we consider three perspectives, enrichment of reactants, more effective diffusion and reactions, and stronger electronic interactions at angstrom-scales, which have been theoretically-proposed to be the potential main contributors to the enhanced angstrom-confined catalytic performance. We have significantly strengthened the mechanism part by rephrasing and adding more discussions. Meanwhile, we have also supplemented molecular dynamics (MD) simulations to illustrate the transport properties of PMS, ranitidine, and water within the interlayer of Co-TiO_x nanosheets. Herein, the mechanism of the Co-TiO_x membrane-based angstrom-confined catalysis activated with PMS was investigated using reactive oxygen species (ROS) scavenging experiments, electron paramagnetic resonance (EPR), DFT and MD.

These have been supplemented in the revised manuscript as shown below.

“Towards understanding the angstrom-confinement catalysis, enrichment of reactants, more effective diffusion and reactions, and stronger electronic interactions at angstrom-scales have been theoretically-proposed to be the potential main contributors to the enhanced catalytic performance^{2,10}. As such, the mechanism of the Co-TiO_x membrane-based angstrom-confined catalysis activated with PMS was investigated using ROS scavenging experiments, electron paramagnetic resonance (EPR), DFT and molecular dynamics (MD).” (Page 15, Line 280–286)

“..... Compared with the batch suspension system, the Co-TiO_x membrane would effectively enrich these ¹O₂, SO₄^{•-} and •OH radicals within its numerous 4.6 Å interlayer free spacings, and thus significantly enhancing the angstrom-confined catalytic performance.” (Page 16, Line 318–321)

“..... For hydrated-state Co-TiO_x membrane that contained numerous angstrom-confined channels of 4.6 Å, PMS dissociation occurred through the cleavage of S–O bond with the subsequent generation of ¹O₂. However, that was not the case at the interlayer free spacings of 8 and 50 Å, where the S–O bond was stretched but not cleaved. As such, theoretically, there will be more ¹O₂ generated in the 4.6 Å interlayer free spacings of Co-TiO_x membrane compared to the cases with 8 and 50 Å interlayer free spacings, contributing to the enhanced catalytic performance.” (Page 17, Line 329–333)

“.....However, much weaker electrostatic interactions were found at the larger interlayer free spacings of 8 and 50 Å compared to the case with 4.6 Å interlayer free spacings (Fig. 4C). The adsorption energy (E_{ads}) of PMS increased from -0.82 to -3.96 eV when the free interlayer

spacing decreased from 50 to 4.6 Å. Hence, the angstrom-confined spaces within the Co-TiO_x membrane can significantly enhance the E_{ads} of PMS and the radical yields, facilitating the angstrom-confined catalytic performance.” (Page 18, Line 342–347)

“In addition, the angstrom-scaled interlayer spacing within Co-TiO_x membrane can greatly enhance the interaction between the ROS and target pollutants with more effective diffusion and reactions, which can also significantly accelerate the catalytic process^{2,10}. This is indeed the case as confirmed by the MD results. MD simulations were performed to illustrate the transport properties of PMS, ranitidine, and water within the interlayer of Co-TiO_x nanosheets. Note that the MD process with the solvent module having 1 ranitidine molecule, 5 PMS molecules and 200 water molecules requires the minimal interlayer free spacing of Co-TiO_x of 2.5 nm. As such, three different interlayer-free spacings (2.5, 3.5, 5 nm) of Co-TiO_x were investigated by MD to compare the mass transfer performance. As shown in Fig. S15, the diffusion of PMS, ranitidine, and H₂O molecules to the surface of Co-TiO_x nanosheets can be much easier with the greatly shortened migration distance as evidenced by the Mean Square Displacement (MSD) curve with the decreasing interlayer-free spacing. As such, it can be expected that the concentration of PMS and ranitidine molecules on the surface of Co-TiO_x nanosheets would be much higher at smaller interlayer-free spacings especially at angstrom-scales, which could enhance the effective collision of ROS and target pollutants, maximize the utilization of reactive radicals and thus significantly promote the catalytic degradation reactions. Moreover, the hydrogen bonding interaction between water molecules and Co-TiO_x nanosheets becomes much stronger at smaller interlayer spacings especially at 4.6 Å (Fig. S16), which could also effectively facilitate the diffusion process within the angstrom-scale nanochannels and thus fast the catalytic

reactions^{2,10}.” (Page 18-19, Line 361–382)

Fig. S15. MD simulation snapshot of the diffusion of PMS, ranitidine, and H₂O molecules (A-C) and the MSD of PMS molecules (D) and ranitidine molecules (E) inside Co-TiO_x nanochannels with three different interlayer free spacings (2.5, 3.5, 5 nm).

Fig. S16. The hydrogen bonding interaction between water molecules and Co-TiO_x nanosheets with different interlayer free spacings (0.46, 2.5, 3.5, 5 nm).

Comment 3: Some minor comments: L 157, 3.67? and 3.76? typo for further check; Fig. 3D, it seems unnecessary for such comparison in scientific viewpoint.

Response: Thanks. We have corrected the typo in the revised manuscript as shown below.

“...there was an increase in the d -spacing from 3.67 Å to 3.76 Å...”

As for Fig. 3D, the k value for the Co-TiO_x membrane/PMS system is 63600 min⁻¹, which is 5–7 orders of magnitude higher than those of the conventional heterogeneous catalytic systems (0.0032–0.33 min⁻¹). We list all values at different conditions in Table S1 for comparison. Herein, we aim to highlight the unprecedented reaction rate within 4.6 Å channels of Co-TiO_x membrane and think figure could be more direct and obvious to present the huge catalytic difference between the angstrom-confined Co-TiO_x membrane and the conventional non-confined catalytic systems.

REVIEWER COMMENTS

Reviewer #1 (Remarks to the Author):

The revised manuscript answered all the queries. Based on these answers I agree with its publication in the revised form.

Reviewer #3 (Remarks to the Author):

The authors have addressed my earlier comments carefully. After further consideration, I have a new comment on the work, i.e., does the oxidation occur inside the nanoconfined space? If the reaction space is confined at an Angstrom level, how does the target compound (of size larger than the size particularly in water) diffuse inside? In addition, in Fig 3B, a better performance was observed for the membrane operation mode than in the batch mode. Does it result from the reaction between the bulk sheets instead of the angstrom-size space? I think the authors should provide the solid evidence to verify the above assumption before possible publication in NC.

Manuscript ID: NCOMMS-22-07845A

**Angstrom-confined catalytic water purification within Co-TiO_x laminar membrane
nanochannels**

REVIEWER 3

General Comment: The authors have addressed my earlier comments carefully. After further consideration, I have a new comment on the work. I think the authors should provide the solid evidence to verify the above assumption before possible publication in NC.

Response: Thanks for your additional comments to further improve this manuscript. A point-by-point response on all issues raised is provided below.

Comment 1: Does the oxidation occur inside the nanoconfined space? If the reaction space is confined at an Angstrom level, how does the target compound (of size larger than the size particularly in water) diffuse inside? In addition, in Fig 3B, a better performance was observed for the membrane operation mode than in the batch mode. Does it result from the reaction between the bulk sheets instead of the angstrom-size space?

Response: Thanks for your comments. Herein, please note that the loading amount of Co-TiO_x nanosheets on the membrane surface for the membrane operation mode and the mass of bulk Co-TiO_x nanosheets for the batch mode are the same in Fig. 3B, however, they delivered a huge difference in catalytic performance (5.5 ms for the 100% ranitidine removal-membrane operation mode vs 20 min for the 85.4% ranitidine removal-batch mode). The significantly improved catalytic performance for the membrane operation mode is mainly resulted from the angstrom-

confinement catalysis within laminar membrane nanochannels. The enrichment of reactants, more effective diffusion and reactions, and stronger electronic interactions at angstrom-scales are the main contributors to the enhanced catalytic performance. We have supplemented ab initio molecular dynamics (AIMD) and molecular dynamics (MD) results to further elucidate the diffusion process in the 4.6 Å interlayer nanochannels of Co-TiO_x membrane.

The diffusion process of target compounds within membrane nanochannels can be divided into two processes, diffusion from aqueous solution into membrane nanochannels and diffusion within membrane nanochannels. 1) According to DFT results (Fig. S8), ranitidine, PMS, and water can enter the 4.6 Å interlayer nanochannels of Co-TiO_x membrane. The larger size ranitidine molecules are more prone to deform, rotate, bend and twist when transporting into the 4.6 Å interlayer nanochannels of Co-TiO_x membrane (Nat. Nanotechnol., 2021, 16, 989-995). The molecular sizes of PMS, ranitidine and water within membrane nanochannels are 1.078, 2.659 and 0.627 Å, respectively (Fig. S8). This is also confirmed by the AIMD results that ranitidine, PMS, and water can enter the 4.6 Å Co-TiO_x membrane nanochannels as shown in Fig. S15A and C. In addition, the diffusion of PMS and ranitidine to the surface of Co-TiO_x nanosheets can be much easier with the greatly shortened migration distance as evidenced by the Mean Square Displacement (MSD) curves as shown in Fig. S16A. As such, it can be expected that the concentration of PMS and ranitidine molecules on the surface of Co-TiO_x nanosheets would be much higher at smaller interlayer-free spacings especially at angstrom-scales (Fig. S17), which could enhance the effective collision of ROS and target pollutants, maximize the utilization of reactive radicals and thus significantly promote the catalytic degradation reactions. This is indeed the case as evidenced by the AIMD results that only after 1500 fs, the PMS and

ranitidine molecules decompose into various fragments in the 4.6 Å interlayer nanochannels of Co-TiO_x membrane (Fig. S15B and D). 2) As such, the diffusion process of PMS fragments, ranitidine fragments, and water molecules inside the 4.6 Å Co-TiO_x membrane nanochannels was further investigated. As shown in Fig. S16B, the diffusion process of PMS fragments, ranitidine fragments, and water molecules follows the order: water molecules > ranitidine fragments > PMS fragments. MD simulations further show that water molecules as the form of monolayer water molecules rapidly move through the 4.6 Å Co-TiO_x interlayer nanochannels by the driving force of hydrogen bonding (Fig. S18). Water molecules with a highly ordered structure can significantly accelerate the diffusion process at angstrom-scales (Science 2014, 343, 752–754; Science 2012, 335, 442–444) and thus facilitate the catalytic reactions. Moreover, the hydrogen bonding interaction between water molecules and Co-TiO_x nanosheets becomes much stronger at smaller interlayer spacings especially at 4.6 Å (Fig. S19), which could also effectively facilitate the diffusion process within the angstrom-scale nanochannels and thus fast the catalytic reactions (J. Am. Chem. Soc. 2015, 137, 477–482; Acc. Chem. Res. 2011, 44, 553–562).

We have supplemented these contents as well as figures in the revised manuscript as shown below. (Page 18-20, Line 361-402)

“In addition, the angstrom-scaled interlayer spacing within Co-TiO_x membrane can greatly enhance the interaction between the ROS and target pollutants with more effective diffusion and reactions, which can also significantly accelerate the catalytic process^{2,10}. This is indeed the case as confirmed by the AIMD and MD results. AIMD and MD simulations were performed to

further elucidate the diffusion process of PMS, ranitidine, and water in the 4.6 Å interlayer nanochannels of Co-TiO_x membrane.

The diffusion process of PMS, ranitidine, and water within membrane nanochannels can be divided into two processes, diffusion from aqueous solution into membrane nanochannels and diffusion within membrane nanochannels. According to DFT results (Fig. S8), ranitidine, PMS, and water can enter the 4.6 Å interlayer nanochannels of Co-TiO_x membrane. The larger size ranitidine molecules are more prone to deform, rotate, bend and twist when transporting into the 4.6 Å interlayer nanochannels of Co-TiO_x membrane²⁷. The molecular sizes of PMS, ranitidine and water within membrane nanochannels are 1.078, 2.659 and 0.627 Å, respectively (Fig. S8). This is also confirmed by the AIMD results that ranitidine, PMS, and water can enter the 4.6 Å Co-TiO_x membrane nanochannels as shown in Fig. S15A and C. In addition, the diffusion of PMS and ranitidine to the surface of Co-TiO_x nanosheets can be much easier with the greatly shortened migration distance as evidenced by the Mean Square Displacement (MSD) curves as shown in Fig. S16A. As such, it can be expected that the concentration of PMS and ranitidine molecules on the surface of Co-TiO_x nanosheets would be much higher at smaller interlayer-free spacings especially at angstrom-scales (Fig. S17), which could enhance the effective collision of ROS and target pollutants, maximize the utilization of reactive radicals and thus significantly promote the catalytic degradation reactions. This is indeed the case as evidenced by the AIMD results that only after 1500 fs, the PMS and ranitidine molecules decompose into various fragments in the 4.6 Å interlayer nanochannels of Co-TiO_x membrane (Fig. S15B and D). As such, the diffusion process of PMS fragments, ranitidine fragments, and water molecules inside the 4.6 Å Co-TiO_x membrane nanochannels was further investigated. As shown in Fig. S16B, the

diffusion process of PMS fragments, ranitidine fragments, and water molecules follows the order: water molecules > ranitidine fragments > PMS fragments. MD simulations further show that water molecules as the form of monolayer water molecules rapidly move through the 4.6 Å Co-TiO_x interlayer nanochannels by the driving force of hydrogen bonding (Fig. S18). Water molecules with a highly ordered structure can significantly accelerate the diffusion process at angstrom-scales^{17,18} and thus facilitate the catalytic reactions. Moreover, the hydrogen bonding interaction between water molecules and Co-TiO_x nanosheets becomes much stronger at smaller interlayer spacings especially at 4.6 Å (Fig. S19), which could also effectively facilitate the diffusion process within the angstrom-scale nanochannels and thus fast the catalytic reactions^{2,10}.”

Fig. S15. AIMD simulations (A and C: initial state; B and D: final state after 1500 fs) of the diffusion of PMS, ranitidine, and H₂O molecules inside Co-TiO_x nanochannels with the interlayer free spacing of 0.46 nm.

Fig. S16. MSD curves of PMS, ranitidine, and H₂O molecules (A) and PMS fragments,

ranitidine fragments, and H₂O molecules (B) inside Co-TiO_x nanochannels with the interlayer free spacing of 0.46 nm.

Fig. S18. The transportation of water molecules within Co-TiO_x nanosheets and hydrogen bonding interaction between water molecules and Co-TiO_x nanosheets with the interlayer free spacing of 0.46 nm.

REVIEWERS' COMMENTS

Reviewer #3 (Remarks to the Author):

The authors have carefully addressed my new comments and I think it is publishable now.

Response to reviewers' comments and changes made in the manuscript

Manuscript ID: NCOMMS-22-07845B

**Angstrom-confined catalytic water purification within Co-TiO_x laminar membrane
nanochannels**

REVIEWER 3

Comment: The authors have carefully addressed my new comments and I think it is publishable now.

Response: Thank you for your recognition of this work and your suggestions on this manuscript.